# Hidden Markov modeling for maximum probability neuron reconstruction

Thomas L. Athey [1,2 ✉], Daniel J. Tward [3,4 ✉], Ulrich Mueller[5 ✉], Joshua T. Vogelstein [1,2,6,7 ✉] & Michael I. Miller[1,2,6,7 ✉]

Recent advances in brain clearing and imaging have made it possible to image entire mammalian brains at sub-micron resolution. These images offer the potential to assemble brain-wide atlases of neuron morphology, but manual neuron reconstruction remains a bottleneck. Several automatic reconstruction algorithms exist, but most focus on single neuron images. In this paper, we present a probabilistic reconstruction method, ViterBrain, which combines a hidden Markov state process that encodes neuron geometry with a random field appearance model of neuron fluorescence. ViterBrain utilizes dynamic programming to compute the global maximizer of what we call the most probable neuron path. We applied our algorithm to imperfect image segmentations, and showed that it can follow axons in the presence of noise or nearby neurons. We also provide an interactive framework where users can trace neurons by fixing start and endpoints. ViterBrain is available in our open-source Python package `brainlit`.

[1] Department of Biomedical Engineering, Johns Hopkins University, Baltimore, MD, USA. [2] Institute of Computational Medicine, Johns Hopkins University, Baltimore, MD, USA. [3] Department of Computational Medicine, University of California at Los Angeles, Los Angeles, CA, USA. [4] Department of Neurology, University of California at Los Angeles, Los Angeles, CA, USA. [5] Department of Neuroscience, Johns Hopkins University, Baltimore, MD, USA. [6] Center for Imaging Science, Johns Hopkins University, Baltimore, MD, USA. [7] Kavli Neuroscience Discovery Institute, Johns Hopkins University, Baltimore, MD, USA. ✉email: tathey1@jhu.edu; DTward@mednet.ucla.edu; umuelle3@jhmi.edu; jovo@jhu.edu; mim@jhu.edu

Neuron morphology has been a central topic in neuroscience for over a century, as it is the substrate for neural connectivity, and serves as a useful basis for neuron classification. Technological advances in brain clearing and imaging have allowed scientists to probe neurons that extend throughout the brain, and branch hundreds of times[1]. It is becoming feasible to assemble a brainwide atlas of cell types in the mammalian brain which would serve as a foundation for understanding how the brain operates as an integrated circuit, or how it fails in neurological disease. One of the main bottlenecks in assembling such an atlas is the manual labor involved in neuron reconstruction.

In an effort to accelerate reconstruction, many automated reconstruction algorithms have been proposed, especially over the last decade. In 2010, the DIADEM project brought multiple institutions together to consolidate existing algorithms, and stimulate further progress by generating open access image datasets, and organizing a contest for reconstruction algorithms[2]. Several years later, the BigNeuron project continued the legacy of DIADEM, this time establishing a common software platform, Vaa3D, on which many of the state of the art algorithms were implemented[3]. Acciai et al.[4] offers a review of notable reconstruction algorithms up to, and through, the BigNeuron project.

Previous approaches to automated neuron reconstruction have used shortest path/geodesic computation[5–7], minimum spanning trees[8], Bayesian estimation[9], tracking[10,11], and deep learning[12–14]. Methods have also been developed to enhance, or extend existing reconstruction algorithms[15–18]. Also, some works focus on the subproblem of resolving different neuronal processes that pass by closely to each other[19–21].

Both DIADEM and BigNeuron initiatives focused on the task of single neuron reconstruction so most associated algorithms fail when applied to images with several neurons. However, robust reconstruction in multiple neuron images is essential in order to assemble brainwide atlases of neuron morphology.

We propose a probabilistic model-based algorithm, ViterBrain, that operates on imperfect image segmentations to efficiently reconstruct neuronal processes. Our estimation method does not assume that the image outside the reconstruction is background, and thus allows for the existence of other neurons. Our approach draws upon two major subfields in Computer Vision, appearance modeling and hidden Markov models, and generates globally optimal solutions using dynamic programming. The states of our model are locally connected segments. We score the state transitions using appearance models such as exhibited by Kass and Cohen's early works[22,23] on active shape modeling and their subsequent application by Wang et al.[6] for neuron reconstruction.

For our own approach, we exploit foreground-background models of image intensity for the data likelihood term in the hidden-Markov structure. We quantify the image data using and intensity autocorrelation and kernel density estimates in order to validate our model assumptions.

Our probabilistic models are hidden Markov random fields, but we reduce the computational structure to a hidden Markov model (HMM) since the latent axonal structures have an absolute ordering. Hidden Markov modeling (HMM) involves two sequences of variables, one is observed and one is hidden. A popular application of HMM's is in speech recognition where the observed sequence is an audio signal, and the hidden variables a sequence of words[24]. In our setting, the observed data is the image, and the hidden data is the contour representation of the axon or dendrite's path.

The key advantages to using HMMs in this context are, first, that neuronal geometry can be explicitly encoded in the state transition distribution. We utilize the Frenet representation of curvature in our transition distribution, which we have studied previously in Athey et al.[25], Khaneja et al.[26]. Secondly, globally optimal estimates can be computed efficiently using dynamic programming in HMMs[26,27]. The well-known Viterbi algorithm computes the MAP estimate of the hidden sequence in an HMM. Our approach, inspired by the Viterbi algorithm, also computes globally optimal estimates. Thus, our optimization method is not susceptible to local optima that exist in filtering methods, or gradient methods in active shape modeling.

In this work, we apply our hidden Markov modeling framework to the output of low-level image segmentation models. Convolutional neural networks have shown impressive results in image segmentation[28], but it only takes a few false negatives to sever neuronal processes that are often as thin as one micron (Fig. 1). Our method strings together the locally connected components of the binary image masks into a reconstruction with a global ordering. Thus, our method is modular enough to leverage state of the art methods in machine learning for image segmentation.

We apply our method to data from the MouseLight project at Janelia Research Campus[1], and focus on the endpoint control problem in a single neuronal process i.e. start and end points are fixed. We introduce the use of Frechet distance to quantify the precision of reconstructions and show that our method has comparable precision to state of the art, when the algorithms are successful.

## Results

**Overview of ViterBrain.** Viterbrain takes in an image, and associated neuron mask produced by some image segmentation

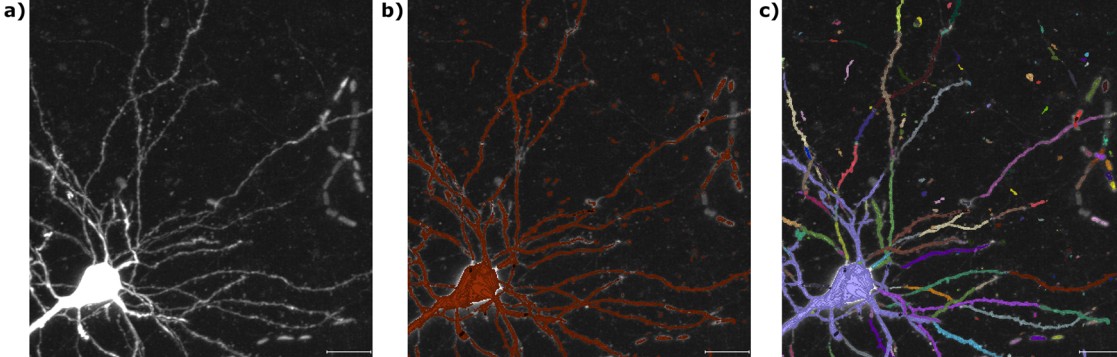

**Fig. 1 Image segmentation models sever neuronal processes. a** An image subvolume from the MouseLight project containing a single neuron. **b** The same image overlaid with a binary image mask in brown. This mask was generated by the random forest based software Ilastik[32] and illustrates the typical output of an image segmentation model. **c** The same binary image mask, with a different color for each connected component. The variety of colors shows that the neuron has been severed into several pieces. All panels are maximum intensity projections (MIPs), and the scale bar represents 15 microns.

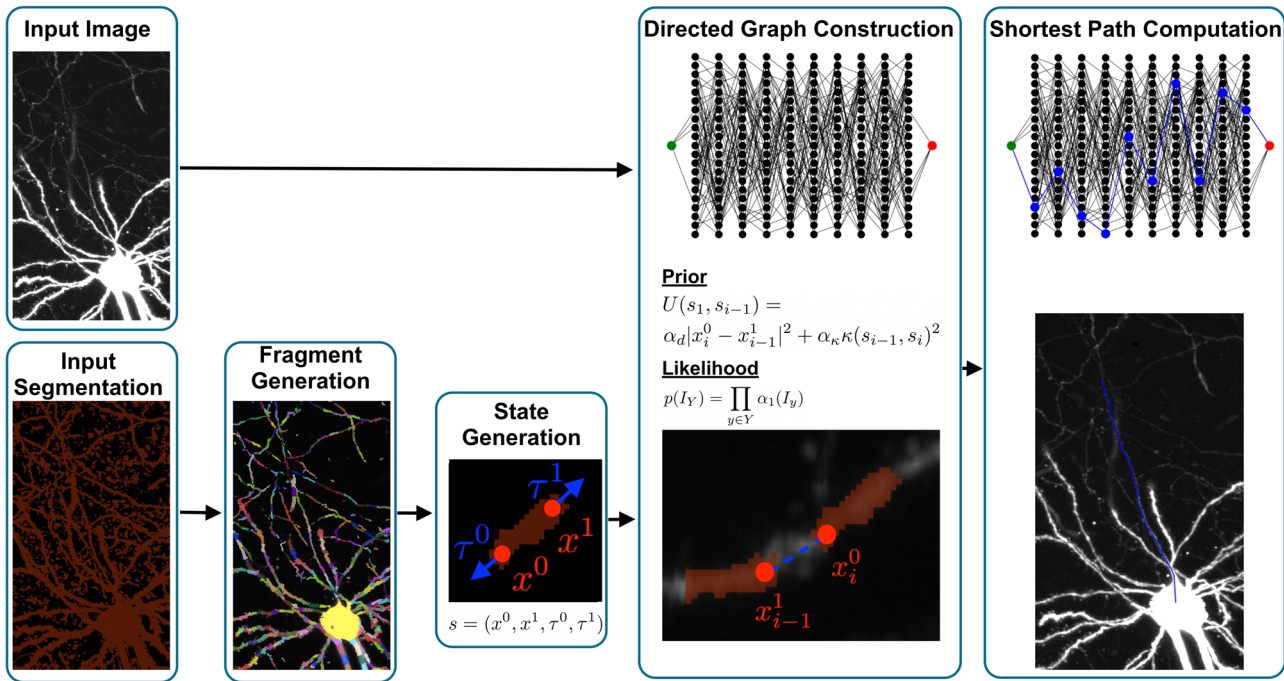

**Fig. 2 Summary of the ViterBrain algorithm.** The algorithm takes in an image and a binary mask that might have severed, or fused neuronal processes. First, the mask is processed into a set of fragments. For each fragment, the endpoints ($x^0$, $x^1$) and endpoint orientations ($\tau^0$, $\tau^1$) are estimated and added to the state space. Next, transition probabilities are computed from both the image and state data to generate a directed graph reminiscent of the trellis graph in classic hidden Markov modeling. The transition prior depends on spatial distance between fragments, $|x_i^0 - x_{i-1}^1|$, and curvature of the path that connects them, $\kappa(s_{i-1}, s_i)$, and these two terms are balanced by the hyperparameters $\alpha_d$, $\alpha_\kappa$. The transition likelihood depends on the local image intensity $\alpha_1(I_y)$. Finally, a shortest path algorithm is applied to compute the maximally probable state sequence connecting the start to the end state.

model. The algorithm starts by processing the mask into neuron fragments and estimating fragment endpoints and orientation to generate the states. Next, both the prior and likelihood terms of the transition probabilities are computed to construct a directed graph reminiscent of the trellis from the HMM work in Forney[27]. Lastly, the shortest path between states is computed with dynamic programming. We have proven that the shortest path is the most probable estimate state sequence formed by a neuronal process (Statement 1). An overview of our algorithm is shown in Fig. 2 and details are given in the Methods. We validated our algorithm on subvolumes of one of the MouseLight whole-brain images[1]. The image was acquired via serial two-photon tomography at a resolution of 0.3μm × 0.3μm × 1μm per voxel. Viterbrain is available in our open source Python package brainlit: http://brainlit.neurodata.io/.

**Modeling image intensity**. Figure 3a shows the correlations of image intensities between voxels at varying distances of separation. As is typical for natural images, voxels that are close by each other have positively correlated intensities, and those farther away are uncorrelated. In the case of foreground voxels, correlations become weak beyond a distance of about 10 microns, with background voxel correlation decaying rapidly. This lends support to our assumption that voxel intensities are conditionally independent processes, conditioned on the foreground/background model (Eq. (3b)). This assumption is one of the central features of our model because it provides for computational tractability.

Figure 3b shows kernel density estimates (KDEs) of the foreground and background image intensity distributions. The distributions vary greatly between the three image subvolumes, implying that modeling the image process as homogeneous throughout the whole brain would be inappropriate. Additionally,

the distributions do not appear to be either Gaussian or Poisson. Indeed, Kolmogorov–Smirnov tests rejected the null hypothesis for both Gaussian and Poisson goodness of fit in all cases, with all p-values below $10^{-16}$. For that reason, we exploit the independent increments properties of Poisson emission conditioned on the underlying intensity model, but do not assume that the marginal probabilities are Poisson (or Gaussian), instead, we estimate the intensity distributions from the data itself using KDEs (denoted $\alpha_0(\cdot)$, $\alpha_1(\cdot)$ in Section The Bayesian Appearance Imaging Model).

**Maximally probable axon reconstructions**. Figure 4 demonstrates the reconstruction method on both a satellite image of part of the Great Wall of China, and part of an axon. Different image segmentation models were used to generate the fragments in the two cases, but the process of joining fragments into a reconstruction was the same. The algorithm reconstructs the one dimensional structure in both cases.

Since the state transition probabilities are modeled with a Gibbs distribution indexed by the state (giving the Markov property), reconstructions are driven by relative energies of different transitions. Thus, our method can still be successful in the presence of luminance dropout, as long as the neuronal process is relatively isolated (Fig. 5). Our geometric prior has two hyperparameters, $\alpha_d$ and $\alpha_\kappa$, which determine the influence of distance and curvature, respectively, on the probability of connection between two neuronal fragments.

Figure 6a shows various examples of maximally probable reconstructions. The algorithm was run with the same hyperparameters in all cases: $\alpha_d = 10$ and $\alpha_\kappa = 1000$.

In some cases, reconstruction accuracy is sensitive to hyperparameter values (Fig. 6b). Higher values of $\alpha_d$ penalize transitions between fragment states with large gaps; higher values of $\alpha_\kappa$ penalize state transitions with sharp angles as measured by

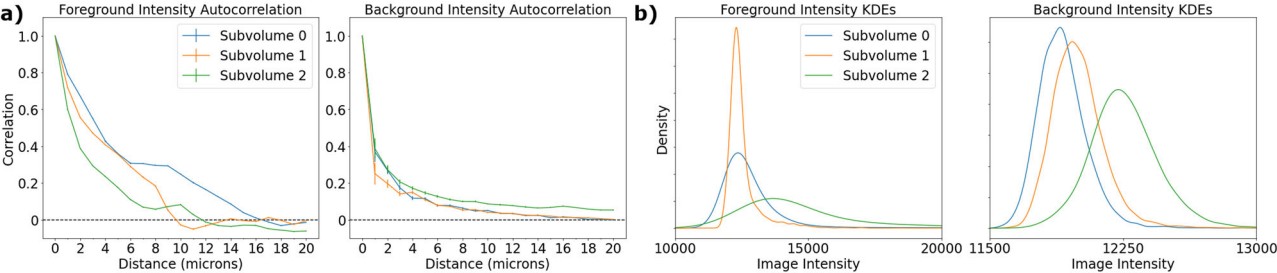

**Fig. 3 Characterization of voxel intensity distributions in three different subvolumes of one of the Mouselight whole-brain images. a** Correlation of intensities between voxels at varying distances from each other. The curves show that intensities are only weakly correlated ($\rho < 0.4$) at a distance of > 10 microns for foreground voxels, or a distance of > 2 microns for background voxels. Error bars represent a single standard deviation of the Fisher z-transformation of the correlation coefficient. Each curve was generated from all pairs of 5000 randomly sampled voxels. **b** Kernel density estimates (KDEs) of foreground and background intensity distributions. A subset of the voxels in each subvolume was manually labeled, then used to train an Ilastik model to classify the remaining voxels. Each KDE was generated from 5000 voxels, according to the Ilastik classifications. KDEs were computed using scipy's Gaussian KDE function with default parameters[38].

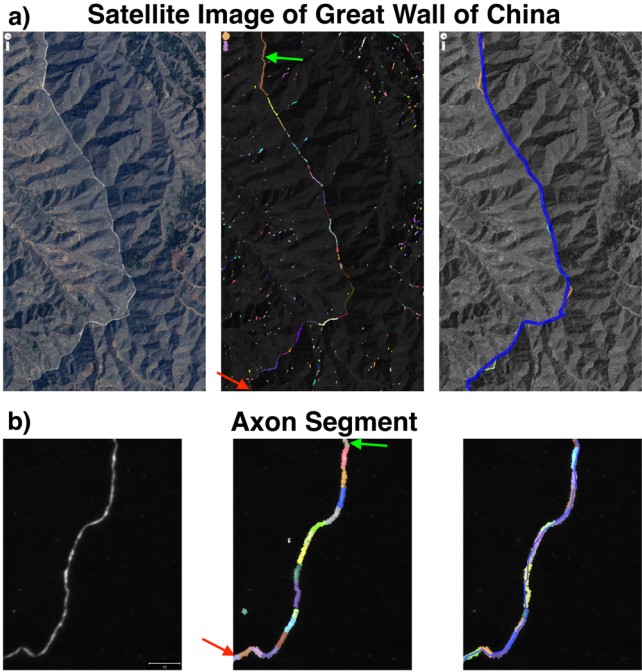

**Fig. 4 Demonstration of maximally probable reconstruction on isolated linear structures. a** A satellite image of part of the Great Wall of China and **b** a neuronal process from the MouseLight dataset (MIP). Left panels show the original images. Middle panels shows the space of fragments, $\mathcal{F}$, pictured in color. The green and red arrows indicate the start and end states of the reconstruction task, respectively. The right panels show the most probable fragment sequences, where the fragments are colored and overlaid with a blue line connecting the endpoints of the fragments. The scale bar in **b** represents 10 microns.

their discrete curvature. It is important to note that the transition distributions depend on the exact values of $\alpha_d$ and $\alpha_\kappa$, not just, for example, the ratio between them. We found $\alpha_d = 10$ and $\alpha_\kappa = 1000$ to be effective for the reconstructions shown throughout the paper, but these values should be adjusted according to the quality of fragments, and the geometry of the neurons being reconstructed.

**Signal dropout causes censored fragment states**. Though our method is robust to fragment dropout in certain contexts, it is less robust when there is dropout near a parallel neuronal process

(Supplementary Fig. 2). Since the reconstruction is ultimately a sequence of fragments, missing fragments forces the algorithm to choose between jumping to a nearby adjacent fragment at the expense of a penalty due to high curvature or continuing on course at the expense of a penalty due to high inter-fragment distance.

The most obvious source of fragment dropout is low image luminance, leading to false negatives in the initial image segmentation. However, the reason for fragment dropout depends on the underlying segmentation model. Empirically, we found that the reconstruction algorithm fails when the fragment generation process neglects portions that are greater than ~10 μm in length.

**Comparison to state of the art**. We examined the accuracy of ViterBrain compared to state of the art reconstruction algorithms. We identified four algorithms that have accompanying publications, and open-source implementations. The first method is APP2 which starts with an oversegmentation of the neuron using a shortest path algorithm, then prunes spurious connections[19]. The second is Snake which is based on active contour modeling[6]. The third is called Advantra, based on the particle filtering approach by Radojevic and Meijering[29]. The final reconstruction software we use is GTree[30], which uses the algorithm outlined in Quan et al.[21]. This algorithm is similar to APP2 in that it starts with an initial reconstruction that spans several neurons, then identifies false connections. APP2, Snake, and Advantra were both used with their default settings and hyperparameters in Vaa3D 3.2 for Mac. GTree version 1.0.4 was used on Linux. For GTree, the binarization threshold was set to 1.0, which was qualitatively identified as a good threshold to capture the neuron. The default soma radius, 3 μm, was used in the soma identification step.

Shown in Fig. 7a are the results of the various methods on a dataset of 35 subvolumes of a MouseLight whole brain image. Each subvolume contains a cell body, and the initial part of its axon that is covered by the first ten points of the Janelia reconstruction. So, the subvolumes vary in size but usually encompass around $10^6$ cubic microns. The algorithms are evaluated on how well they can trace the axon between fixed endpoints (cell body and tenth axon reconstruction point). The outcomes were classified as either successful (if the axon was fully traced), partially successful (if more than half of the axon was reconstructed as evaluated visually), or failures. According to two proportion z-tests the success rate of ViterBrain (11/35) was higher than all other methods at $\alpha = 0.05$. Also, APP2 had a higher success rate (4/35) than Advantra at $\alpha = 0.05$. The success rates for several of the algorithms are discouragingly low, so we

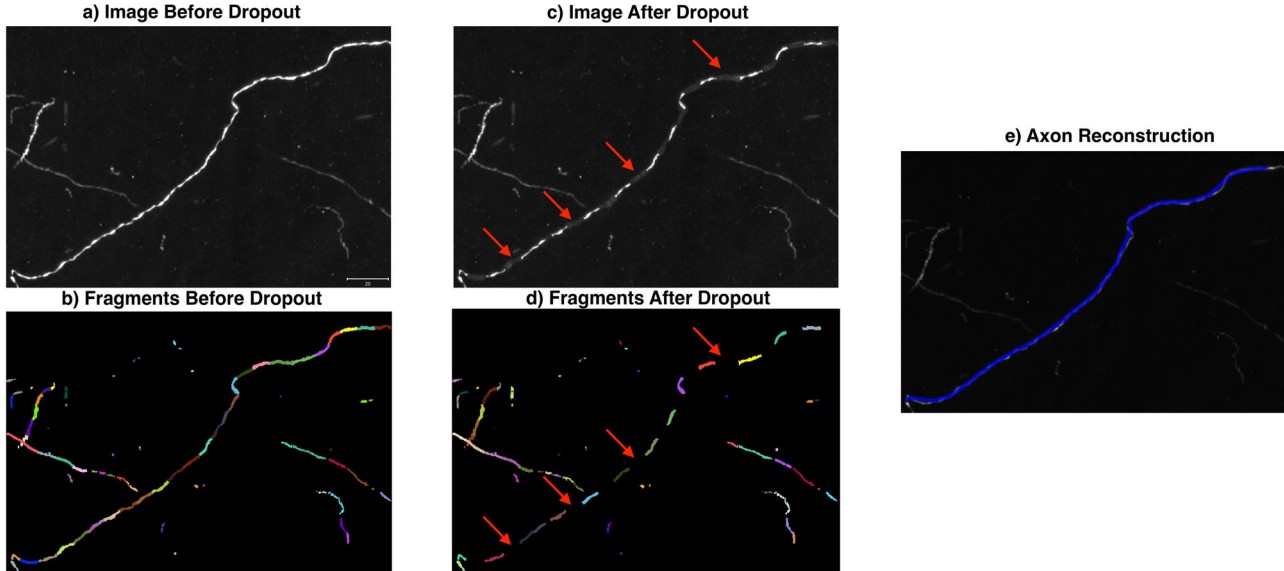

**Fig. 5 ViterBrain is robust to image intensity and fragment dropout when axons are relatively isolated. a** An image subvolume from the MouseLight project containing an axon. The scale bar represents 20 microns. **b** The same image, overlaid with the fragments which are depicted in different colors. **c** The image intensity was censored periodically along an axon path (red arrows). **d** The fragments associated with the censored regions were removed from the fragment space (red arrows). **e** Nonetheless, our algorithm was able to jump over the censored regions to reconstruct this axon. All images are MIPs.

discuss the possible reasons for this in the Discussion, and demonstrate the typical failure modes in Supplementary Figs. 3, 4 and 5.

For successful reconstructions, we examined the precision of the reconstructions using spatial distance (SD) and Frechet distance. We manually upsampled the Janelia reconstructions to provide the most precise ground truth. All reconstructions were sampled at every 1 μm before distances were measured. Spatial distance represents the average distance from a point on one reconstruction to the closest point on the other reconstruction[3]. Frechet distance represents the maximum distance between two reconstructions, and satisfies the criteria to be a mathematical metric and is invariant to reparameterization (see Section Accuracy Metrics for a derivation of Frechet distance). The spatial distance between ViterBrain and manual reconstructions are typically between 1 and 3 microns which is roughly the resolution of the image. Successful reconstructions by the other algorithms are also in this range. Frechet distances are larger, since they indicate maximal distance, not average distance, between curves. We observe that the ~ 5 micron deviations occur most often near the axon hillock where the axon broadens to merge with the soma.

**Proof of concept graphical user interface**. Since our method is formulated as an optimal control problem conditioned on the start and states it is most suited for semi-automatic neuron reconstruction where a tracer could click on two different fragments along a neuronal process, and the algorithm would fill in the gap. The user could proceed to trace several neuronal processes until the full neuron is reconstructed. We designed a proof of concept graphical user interface based on this workflow and show an example set of reconstructions made using our GUI in Fig. 8. The GUI relies on the visualization software napari[31].

## Discussion

This paper presents a hidden Markov model based reconstruction algorithm that connects fragments generated by appearance modeling. Our method converts an image mask into a set of fragments and thus can be applied to the output of an image segmentation model. We chose Ilastik to generate image masks because of its convenient graphical user interface, and high performance on a small number of samples[32]. However, masks could also be generated using a deep learning based model such as Liu et al.[33], Li and Shen[34] or Wang et al.[35].

These fragments are assembled based on the associated appearance model score of the observed image and the discrete numerical curvature and distance of adjacent fragments. In the Methods, we derive the Bayes posterior distribution of the hidden state sequence encoding the axon reconstruction. We also show that applying the polynomial time Viterbi algorithm is not possible for maximum a-posteriori estimation since the state space has to grow to account for possible cycles in the unordered image domain. We address the problem with cycles with a modified procedure for defining the path probabilities making it feasible to efficiently calculate the globally optimal neuron path. The solution for efficiently generating the globally optimal path implies that the local minimum associated to gradient and active appearance models solutions is resolved in this setting.

We apply the algorithm to the fixed endpoint problem in two-photon images of mouse neurons. In a dataset of 35 partial axons, our algorithm successfully reconstructs more axons than the existing state of the art algorithms (Fig. 7). We observed that the most common failure mode in this dataset was when there are extended (>~10 micron) stretches of the putative axonal path where there is significant loss of luminance signals leading to highly censored fragment generation. This implies that the maximally probable HMM procedure is only effective if paired with effective voxel classification tools. The algorithm can also fail in areas densely populated with neuronal processes. We demonstrate that proper selection of the hyperparameters to reflect the density of the fragments and the geometry of the underlying neurons can resolve these issues. Our algorithm is specifically adapted for reconstructing axons in projection neurons in datasets such as MouseLight in two ways. First, the high image quality as indicated by large KL-divergence values (Fig. 3a),

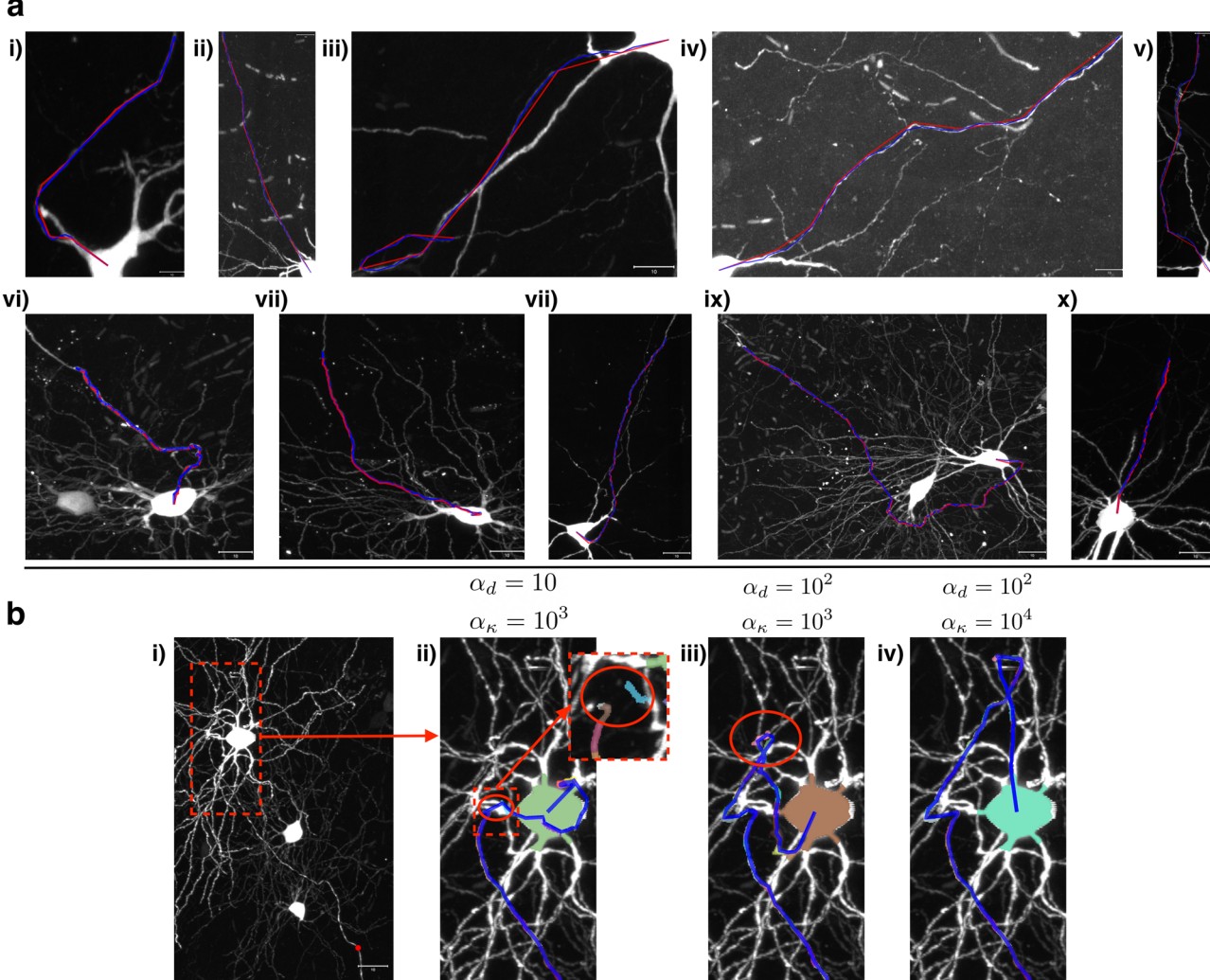

**Fig. 6 Demonstration of ViterBrain. a** Successful axon reconstructions; the ViterBrain reconstructions are shown by the blue line; the manual reconstructions are shown by the red line. The algorithm was run with the same hyperparameters in each case: $\alpha_d = 10$ and $\alpha_\kappa = 1000$. **b** Different hyperparameter values lead to different results. Panel i shows the neuron of interest. Panels ii–iv are close-up views of reconstructions with different hyperparameter values that weigh transition distance ($\alpha_d$) and transition curvature ($\alpha_\kappa$). The red circle in Panel ii indicates where the reconstruction deviated from the true path by jumping ~10$\mu m$ to connect the gray fragment to the light blue fragment. Panel iii shows how a higher $\alpha_d$ value avoids the jump in panel ii, but takes a sharp turn to deviate from the true path (red circle). Finally, in panel iv), the reconstruction avoids both the jump from panel ii) and the sharp turn from panel iii) and follows the true path of the axon back to the cell body. All images are MIPs, and all scale bars represent 10 microns.

makes it straightforward to build an effective foreground-background classifier which is an essential part of our HMM state generation. Secondly, our algorithm encodes the geometric properties of axons such as curvature, which allows our solutions to adapt to the occasional sharp turns in projection axons.

The success rates of the other algorithms on our dataset is quite low, considering the performances that they achieved in their accompanying publications. There are two likely reasons for this, dataset differences, and sub-optimal algorithm settings.

When an algorithm is validated on one type of data, the results do not necessarily hold for datasets of a different type. Since none of the existing algorithms had been designed for the MouseLight data, the unique details of our dataset could lead to reduced performance. For example, several subvolumes in our dataset contain multiple neurons, while two of the algorithms, APP2 and Advantra, were explicitly designed for images containing single neurons. Snake is designed to handle multiple neurons, but is largely validated on DIADEM, a single neuron dataset[2]. Other dataset differences include different image resolutions (or levels of

anisoptropy), different image encodings (8 bit vs. 16 bit), and different signal to noise ratios. Lastly, the existing algorithms are designed to reconstruct all dendrites and axons simultaneously while our task of reconstructing a single section of the axon does not give the algorithms credit for successfully reconstructing other parts of the neuron.

It is also important to note that reconstruction algorithms can be sensitive to hyperparameter settings. All algorithms had different hyperparameter options except for Snake. While we tried various hyperparameter settings for all algorithms, the only non-default setting that clearly improved reconstruction performance was the binarization threshold setting in GTree, all other settings we left to default. It is likely that these settings were not optimal for our dataset, but it is quite time intensive for a typical user to quantitatively determine the optimal settings. Detailed and accessible software documentation makes the process of choosing effective algorithm settings more efficient.

Figures showing the common failure modes for some of the algorithms are shown in Supplementary Figures. Advantra and

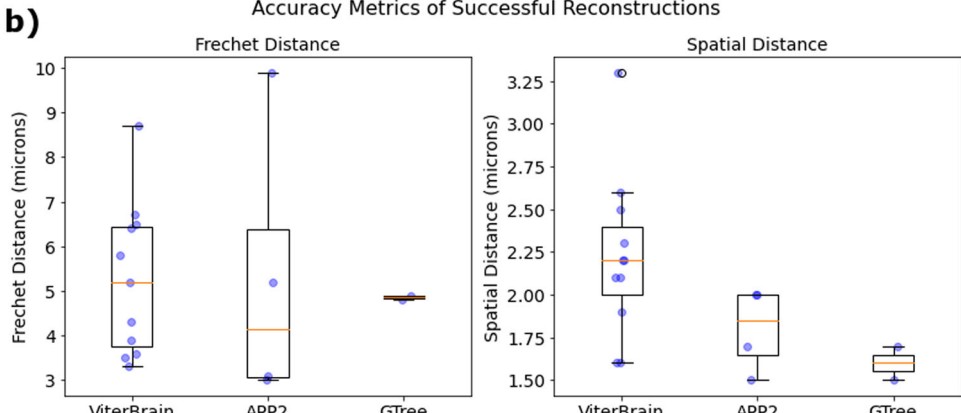

**Fig. 7 Results of reconstruction algorithms on a dataset of 35 subvolumes of a MouseLight whole brain image.** (Snake was only applied to 10 subvolumes due to incoherent results and excessively slow runtimes, see Fig. S4). Each subvolume contained a soma and part of its axon. The task was to reconstruct the portion of the axon that was contained in the image (no branching). First, the algorithms were evaluated visually and classified as successful, partially successful (over half, but not all, of the axon reconstructed), or failed. The table in panel **a** shows these results, along with markers showing statistical significance in a two proportion z-test comparing success rates of the algorithms at $\alpha = 0.05$. For each successful reconstruction, we measured the Frechet distance and spatial distance from the manual ground truth in order to evaluate the precision of the reconstructions. These distances are shown as blue points in **b**, overlaid with standard box and whisker plots (center line, median; box limits, upper and lower quartiles; whiskers, 1.5x interquartile range; points, outliers).

Snake produced incoherent reconstructions on the dataset, and the common failure modes for GTree were early termination of reconstruction, or severing an axon into multiple components. Despite the possible reasons for sub-optimal performance of the other algorithms, we provide evidence that our algorithm is competitive with state-of-the-art methods for reconstructing neuronal processes. Future benchmark comparisons could include reinforcement learning, or recurrent neural network approaches, which have become prevalent in sequential decision processes. However, there is not much scientific literature on these approaches to neuron reconstruction with accompanying functional code.

We have scaled up the ViterBrain pipeline to process images of $3332 \times 3332 \times 1000$ voxels, representing one cubic millimeter of tissue (Supplementary Fig. 6), and we will continue to improve the pipeline until it can be run on whole brain images. The traces generated by our pipeline could also be paired with tools like the one presented in Li et al.[36] which can turn traces into full neuron segmentations complete with axon and dendrite thickness measurements.

The code used in this work is available in our open-source Python package `brainlit`: http://brainlit.neurodata.io/, and a tutorial on how to use the code is located at: http://brainlit.neurodata.io/viterbrain.html.

## Methods

**The Bayesian appearance imaging model**. Our Bayes model is comprised of a prior which models the axons as geometric objects and a likelihood which models the image formation process.

We model the axons as simply connected curves in $\mathbb{R}^3$ written as a function of arclength

$$c(\ell), \ell \geq 0 \,, c(\ell) \in \mathbb{R}^3 \,.$$

We denote the entire axon curve in space as $c(\cdot) := \{c(\ell), \ell \geq 0\}$. To interface the geometric object with the imaging volume we represent the underlying curve $c(\cdot)$ as a delta-dirac impulse train in space. We view the imaging process as the convolution of the delta-dirac impulse train with the point-spread kernel of the imaging platform. We take the point-spread function of the system to be roughly one micron in diameter, implying that the axons are well resolved. The fluorescence process given the axon contour is taken as a relatively narrow path through the imaging domain (~1 μm diameter) with relatively uniform luminance.

We take the image to be defined over the voxel lattice $D = \cup_{i \in \mathbb{Z}^{m^3}} \Delta y_i \subset \mathbb{R}^3$ with centers $y_i \in \Delta y_i$. We model the image as a random field $\{I_{y_i}, \Delta y_i \in D\}$ whose elements are independent when conditioned on the underlying axon geometry, similar to an inhomogeneous Poisson process as described in[37]. We denote the image random field associated to any subset of sites $Y \subset D$, with joint probability conditioned on the axon:

$$I_Y = \{I_{y_i} : \Delta y_i \in Y\} \,; \tag{1a}$$

$$P(I_Y | c(\ell), \ell \geq 0) = \prod_{y \in Y} p(I_y | c(\ell), \ell \geq 0) \,. \tag{1b}$$

Because of the conditional independence, the marginal distribution determines the global joint probabilities. Of course, while the conditional probabilities factor and are conditionally independent, the axon geometry is unknown and the measured image random field is completely connected if the latent axon process is removed. We adopt a two hypothesis formulation $\{f, b\}$ corresponding to a foreground-background model for the images where the marginal probability of a voxel intensity is:

$$P(I_y) = \alpha_1(I_y), y \in \text{foreground} \tag{2a}$$

$$P(I_y) = \alpha_0(I_y), y \in \text{background} \tag{2b}$$

Our conditional independence assumption implies that the joint distribution of group of foreground or group of background voxels can be decomposed into the corresponding marginal distributions over the foreground-background models (2a),(2b); defining the foreground and background sets $Y = Y_f \cup Y_b$, then

$$P(I_Y) = \prod_{y \in Y_f} \alpha_1(I_y) \prod_{y \in Y_b} \alpha_0(I_y) \,. \tag{3a}$$

We define the notations for the joint probability of the set in the foreground for example (or background)

$$\alpha_1(I_{Y_f}) := \prod_{y \in Y_f} \alpha_1(I_y), Y_f \subset \text{foreground.} \tag{3b}$$

Despite the Poisson nature of the image acquisition process, simple scaling or shifting of the imaging data would mean the image intensities are no longer

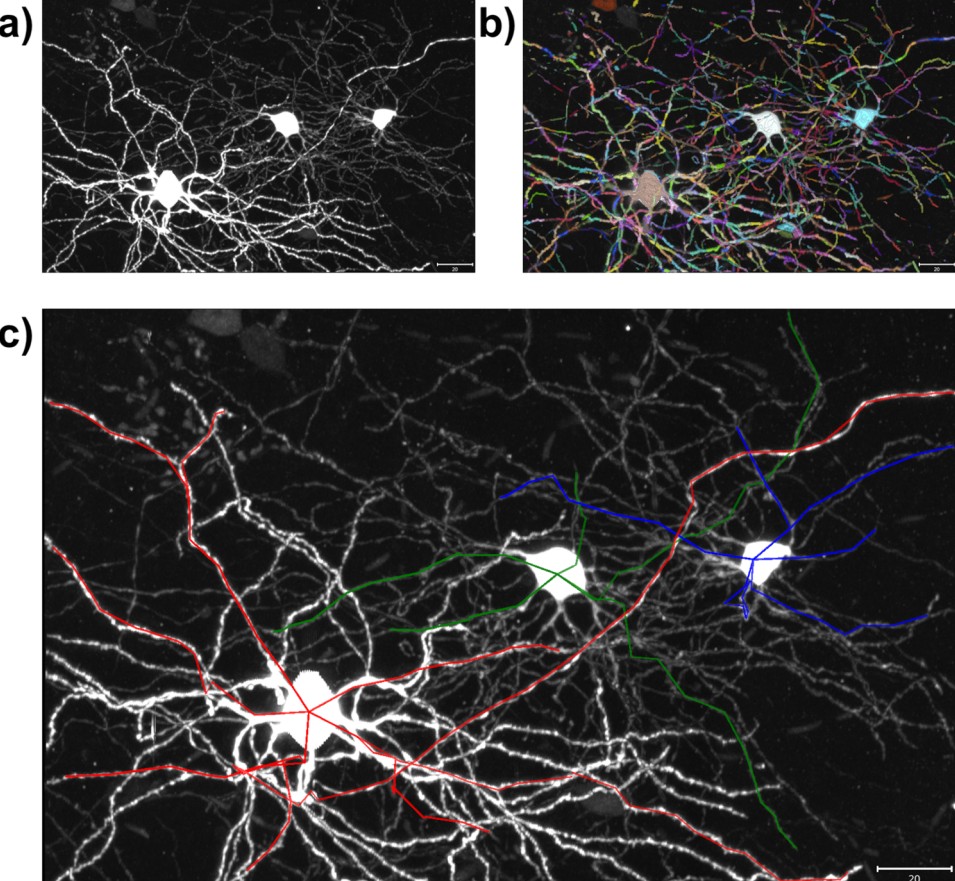

**Fig. 8 Proof of concept graphical user interface. a** Image subvolume presented to the user. **b** Neuron fragments also shown in different colors. The user can then click on two fragments and generate the most probable curve between them. **c** Three partial reconstructions (red, green and blue) of different neurons using the GUI. The scale bars represent 20 microns.

Poisson. To accommodate this effect, we estimate the foreground-background intensity distributions ($\alpha_1(\cdot)$ and $\alpha_0(\cdot)$ respectively) nonparametrically. The simplest nonparametric density estimation technique is using histograms. However, it can be difficult to choose the origin and bin width of histograms, so we opt for a kernel density estimate (KDE) approach. We estimate $\alpha_0(\cdot)$, $\alpha_1(\cdot)$ by labeling a subset of the data as foreground/background then fitting Gaussian KDEs to the labeled data (see Fig. 3b). We use the scipy implementation of Gaussian KDEs[38], with Scott's rule to determine the bandwidth parameter[39]. Under some assumptions on the derivatives of the underlying density, our approach converges to the true density as the number of samples increases (Theorem 6.1 in[39]). Further, the Scott's rule choice for bandwidth is (approximately) optimal with respect to mean integrated square error.

Our approach allows us to empirically estimate the intensity distributions while maintaining the independent increments property of a spatial Poisson process, which is consistent with the autocorrelation curves depicted in Fig. 3a. This is the key property that allows us to factor joint probabilities into products of marginals for sets of voxels.

We note that the foreground-background imaging model allows us to estimate the error rate of classifying a voxel as either foreground or background. In the Neyman-Pearson framework, foreground-background classification is a simple two hypothesis testing problem and the most powerful test at a given type 1 error rate is the log likelihood ratio test. The Kullback–Leibler (KL) divergence between the foreground and background distributions gives the exponential rate at which error rates converge to zero as the number of independent, identically distributed samples increases[40]. In the case of using Gaussians to model the foreground-background distributions, the KL divergence reduces to the squared signal to noise ratio. In the absence of normality, the KL Divergence is the general information theoretic measure of image quality for arbitrary distributions. We propose KL Divergence as an important statistic in evaluating the quality of fluorescent neuron images.

**The prior distribution via Markov state representation on the axons fragments**. Our representation of the observed image is as a hidden Markov random field with the axon as the hidden latent structure. Given the complexity of submicron resolution images, we build an intermediate data structure at the micron

scale that we call fragments $F \subset D$ defined as collections of voxels without any assumed global ordering between them. Each fragment represents a portion of a neuronal process, with a natural orientation given by one end that is closer to the soma. Depicted in Fig. 1 are the fragments shown via different colors. The fragments are a coarser scale voxel and can be viewed analogously as higher order features.

The axon reconstruction problem becomes the reassembly of the fragments along with the imputation of the censored fragments. In the image examples presented in this work, the complexity of the space of fragments is approximately $|\mathcal{F}| = 100{,}000$ for a cubic millimeter of projection neuron image data.

We exploit the computational structures of hidden Markov models (HMMs) when the underlying latent structure is absolutely ordered so that dynamic programming can efficiently compute globally optimal state sequence estimates. From the set of fragments we compute a set of states for the HMM. The states are a simplified, abstract representation of the fragments that contain the minimum information required to specify the HMM. Each state includes endpoints $x^0, x^1 \in \mathbb{R}^3$ in order to compute "gap" or "censored" probabilities, and unit length tangents $\tau^0, \tau^1 \in \mathbb{R}^3$ associated with the endpoints in order to compute curvature. Each fragment generates two states, one for each orientation. The two states are identical except their endpoints are swapped, and their tangents are swapped and reversed. We denote the natural mapping from state to fragment by $F : s \in \mathcal{S} \mapsto F(s) \in \mathcal{F}$.

The collection of states $\mathcal{S} = \{s\}$ is the finite state space of the HMM, and our goal is to estimate the state sequence $(s_1, \ldots, s_n)$ that follows the neuronal process.

Our algorithms exploit two splitting properties, the Markov nature of the state sequence and the splitting of the random field image conditioned on the state sequence. We use the notation $s_{i:j} := (s_i, s_{i+1}, \ldots, s_j)$ for partial state sequences. We model the state sequence $s_1, \ldots, s_n$ as Markov with splitting property:

$$p(s_{i+1:n}, s_{1:i-1}|s_i) = p(s_{i+1:n}|s_i)p(s_{1:i-1}|s_i), \ i = 2, \ldots, n-1,$$

which implies the 1-order Markov property $p(s_i|s_{i-1}, s_{1:i-2}) = p(s_i|s_{i-1})$.

We define the transition probabilities with a Boltzmann distribution with energy $U$:

$$p(s_i|s_{i-1}) = \frac{e^{-U(s_{i-1}, s_i)}}{Z(s_{i-1})}, \ \text{with} \ Z(s_{i-1}) = \sum_{s_i \in \mathcal{S}} e^{-U(s_{i-1}, s_i)},$$

with energy given by

$$U(s_{i-1}, s_i) = \alpha_d |x_i^0 - x_{i-1}^1|^2 + \alpha_\kappa \kappa(s_{i-1}, s_i)^2,$$

where $|\cdot|$ is the standard Euclidean norm. The two hyperparameters $\alpha_d$ and $\alpha_\kappa$, determine the influence of distance and curvature, respectively, on the probability of connection between two neuronal fragments. The term $\kappa(s_{i-1}, s_i)$ approximates the curvature of the path connecting $s_{i-1}$ to $s_i$ as follows:

Define $\tau_c := \frac{x_i^0 - x_{i-1}^1}{||x_i^0 - x_{i-1}^1||}$ which is the normalized vector connecting $s_{i-1}$ to $s_i$. Then we can approximate the squared curvature at $x_{i-1}^1$ and $x_i^0$ with $\kappa_1(s_{i-1}, s_i)^2 = 1 - \tau_{i-1}^1 \cdot \tau_c$ and $\kappa_2(s_{i-1}, s_i)^2 = 1 - \tau_c \cdot (-\tau_i^0)$ respectively. These formulas are derived in Supplementary Method 2 and they approximate curvature as modeled in Athey et al.[25]. Finally, we approximate average squared curvature with the arithmetic mean:

$$\kappa(s_{i-1}, s_i)^2 = \frac{\kappa_1(s_{i-1}, s_i)^2 + \kappa_2(s_{i-1}, s_i)^2}{2}$$

We control the computational complexity associated to computing prior probabilities for all $|\mathcal{S}|^2$ states by restricting the possible state transitions. We set to 0 probability any transitions where the distance between endpoints is greater than 15 $\mu m$ or the angle between states is greater than 150°. We also set the probability that a state transitions to itself to zero.

**Global maximally probable solutions via $O(n|\mathcal{S}|^2)$ calculation**. The maximum a-posteriori (MAP) state sequence is defined as the maximizer of the posterior probability (MAP) over the state sequences $s_{1:n} \in \mathcal{S}^n$, with $|\mathcal{S}|$ finite:

$$\hat{s}_{1:n} := \arg \max_{s_{1:n} \in \mathcal{S}^n} \log p(s_{1:n}|I_D). \qquad (4)$$

The solution space has cardinality $|\mathcal{S}|^n$ so it is infeasible to compute the global maximizer by exhaustive search. Our approach is to rewrite the probability recursively in order to use the Viterbi algorithm and dynamic programming with $O(n|\mathcal{S}|^2)$ time complexity. We rewrite the MAP estimator in terms of the joint probability:

$$\hat{s}_{1:n} = \arg \max_{s_{1:n} \in \mathcal{S}^n} p(s_{1:n}|I_D) = \arg \max_{s_{1:n} \in \mathcal{S}^n} p(s_{1:n}, I_D).$$

The image random field is split or conditionally independent conditioned on the fragment states:

$$p(I_{F(s_i)}, I_{D \setminus F(s_i)}|s_i,) = p(I_{F(s_i)}|s_i)p(I_{D \setminus F(s_i)}|s_i),$$

which implies $p(I_{F(s_i)}|s_i, I_{D \setminus F(s_i)}) = p(I_{F(s_i)}|s_i)$.

Define the indicator function $\delta_A(x) = 1$ for $x \in A$, 0 otherwise.

**Lemma 1**. Defining the shorthand notation identifying fragment sequences with the state sequence:

$$F_{1:n} := (F(s_1), \ldots, F(s_n)),$$

then for $n > 1$ we have the joint probability:

$$p(s_{1:n}, I_D) = \prod_{i=2}^{n} \left( \frac{\alpha_1(I_{F_i})}{\alpha_0(I_{F_i})} \right)^{\delta_{D \setminus F_{1:i-1}}(F_i)} p(s_i|s_{i-1}) p(s_1, I_D). \qquad (5)$$

See Supplementary Method 3 for the proof which rewrites the probability recursively giving the factorization. The proof is similar to the classic HMM decomposition in how it uses the two splitting properties, but there are two differences. The first is that the probability needs to account for the full image, including areas outside of the axon estimate, which explains the presence of both $\alpha_1$ and $\alpha_0$ in Eq. (5). The second difference is that if the state sequence $s_{1:n}$ contains repeated states, then the corresponding image data should not be double counted in the probability. This is enforced by the delta function $\delta(\cdot)$.

It is natural to take the negative logarithm of Eq. (5) to obtain a sum that represents and the cost of a path through a directed trellis graph[27]. Several algorithms exist that solve the shortest path problem in $O(n|\mathcal{S}|^2)$ complexity. However, we cannot use these algorithms directly because the cost function is not sequentially-additive due to the dependence of the indicator function on previous states in the sequence. In Supplementary Method 4, we offer a example demonstrating that directly applying the Viterbi algorithm to this problem does not generate the MAP estimate.

We adjust our probabilistic representation on the $|\mathcal{S}|^n$ paths in order to utilize shortest path algorithms such as Bellman-Ford or Dijkstra's[41,42]. For this we note that the $\frac{\alpha_1(I_{F_i})}{\alpha_0(I_{F_i})}$ term in Eq. (5) may often be greater than 1. In the directed graph formulation (negative logarithm transformation), this can lead to negative cycles in the graph of states. When negative cycles exist, the shortest path problem is ill-posed. To avoid this phenomena we remove the background component of the image from the joint probability, which converts $\frac{\alpha_1(I_{F_i})}{\alpha_0(I_{F_i})}$ to $\alpha_1(I_{F_i})$, and converts our global posterior probability to our path probability formulation.

**Statement 1**. *Define the most probable solution $s_{1:n} \in \mathcal{S}^n$ by the joint probability* $\arg \max_{s_{1:n} \in \mathcal{S}^n} p(s_{1:n}, I_{F_{1:n}})$. *Then we have*

$$\max_{s_{1:n} \in \mathcal{S}^n} p(s_{1:n}, I_{F_{1:n}}) = \max_{s_{1:n} \in \mathcal{S}^n} \prod_{i=2}^{n} \left( \alpha_1(I_{F_i}) \right)^{\delta_{D \setminus F_{1:i-1}}(F_i)} p(s_i|s_{i-1}) p(s_1, I_{F_1}). \qquad (6)$$

*Further, if $\alpha_1(I_y) \leq 1$ for all $y$, then the globally optimal solution to the fixed start and end point problem is a nonrepeating state sequence and can be obtained by computing the shortest path in a directed graph where the vertices are the states, and the edge weight from state $s_{i-1}$ to $s_i$ is given by:*

$$e(s_{i-1}, s_i) = -\log \alpha_1(I_{F_i}) - \log p(s_i|s_{i-1}) \qquad (7)$$

See Supplementary Method 3 for proof. Our reconstruction problem has now become a shortest path problem, and can be solved using one of the several dynamic programming algorithms.

We note that since the path $(s_{1:n})$ defines the subset of the image in the joint probability $(I_{F_{1:n}})$ we can define the probability as a function of only the state sequence $\bar{p}(s_{1:n}) := p(s_{1:n}, I_{F_{1:n}})$ emphasizing that we are solving the most probable path problem.

**Implementation**

*Fragment generation*. Fragments are collections of voxels, or supervoxels, and can be viewed analogously as higher order features such as edgelets or corners. As described in Section The Prior Distribution via Markov State Representation on the Axons Fragments, identifying the subset of fragments that compose the axon, then ordering them becomes equivalent to reconstructing the axon contour model.

The first step of fragment generation is obtaining a foreground-background mask, which could be obtained, for example, from a neural network, or by simple thresholding. In this work, we use an Ilastik model that was trained on three image subvolumes, each of which has three slices that were labeled[32]. During prediction, the probability predictions from Ilastik are thresholded at 0.9, a conservative threshold that keeps the number of false positives low.

The connected components of the thresholded image are split into fragments of similar size by identifying the voxel $v$ with the largest predicted foreground probability and placing a ball $B_v$ with radius 7 $\mu m$ on that voxel. The voxels within $B_v$ are removed and the process is repeated until the component is covered. The component is then split up into pieces by assigning each voxel to the center point from the previous step, $v$, that is closest to it. This procedure ensures that each fragment is no larger than a ball with radius 7 $\mu m$. At this size, it is reasonable to assume that each fragment is associated with only one axon branch since no fragment is large enough to extensively cover multiple branches.

Next, the endpoints $x^0, x^1$ and tangents $\tau^0, \tau^1$ are computed as described in Supplementary Method 2. Each fragment is simplified to the line segment between its endpoints which is rasterized using the Bresenham algorithm[43]. Briefly, the Bresenham algorithm identifies the image axis along which the line segment has the largest range and samples the line once every voxel unit along that axis. Then, the other coordinates are chosen to minimize the distance from the continuous representation line segment.

*Imputing fragment deletions*. In practice the imaging data may exhibit significant dropouts leading to significant fragment deletions. While computing the likelihood of the image data, we augment the gaps between any pair of connected fragments in $F_1, F_2, \ldots$ by augmenting the sequence with imputed fragments $\bar{F}_1, \bar{F}_1, F_2, \bar{F}_2, \ldots$, with $\bar{F}_i \subset D$ the imputed line of voxels which forms the connection between the pair $F_i, F_{i+1}$. For this define the start and endpoint of each fragment as $x^0(F) \in \mathbb{R}^3, x^1(F) \in \mathbb{R}^3$ with line segment connecting each pair:

$$L_{i,i+1} = \{y : y = ax^1(F_i) + (1 - a)x^0(F_{i+1}), a \in [0, 1]\}.$$

The imputed fragment $\bar{F}_i \subset D$ for each pair $(F_i, F_{i+1})$ is computed by rasterizing $L_{i,i+1}$ with the Bresenham algorithm.

The likelihood of the sequence of fragments augmented by the imputations becomes

$$p(s_{1:n}, I_{F_{1:n}}) = \prod_{i=2}^{n} \alpha_1(I_{F_i})^{\delta_{D \setminus F_{1:i-1}}(F_i)} \alpha_1(I_{\bar{F}_i}) p(s_i|s_{i-1}) p(s_1, I_{F_1})$$

*Initial and endpoint conditions*. We take the initial conditions to represent

$$p(s_1, I_{F_1}) = \pi(s_1) p(I_{F_1}|s_1),$$

with $\pi$ the prior on initial state. For all of our axon reconstructions we specify an axonal fragment as the start state $s_{start}$ and set $\pi(s_1) := \delta_{s_{start}}(s_1)$.

The endpoint conditions are defined via a user specified terminal state $s_{term}$ where the path ends giving the maximization:

$$\max_{s_{1:n} \in \mathcal{S}^n} p(s_{1:n}, I_{F_{1:n}}|s_n = s_{term}).$$

The marginal probability on the terminal state always transitions to itself, so that $p(s_n = s_{term}) = \delta_{s_{term}}(s_n)$. Thus, a state sequence solution of length $n$ may end in multiple repetitions of $s_{term}$, such as

$$s_{1:n} = \{s_1, s_2, ..., s_{n'}, s_{term}, s_{term}, ..., s_{term}\}.$$

**Accuracy metrics**. We applied several state of the art reconstruction algorithms to several neurons in the brain samples from the MouseLight Project from HHMI Janelia[1]. In this dataset, projection neurons were sparsely labeled then imaged with a two-photon microscope at a voxel resolution of $0.3 \times 0.3 \times 1 \mu m$. Each axon reconstruction is generated semi-automatically by two independent annotators. The MouseLight reconstructions are sampled roughly every $10 \mu m$, so in some cases we retraced the axons at a higher sampling frequency in order to obtain more precise accuracy metrics.

We quantified reconstruction accuracy using two metrics, the first of which is Frechet distance. Frechet distance is commonly described in the setting of dog walking, where both the dog and owner are following their own predetermined paths. The Frechet distance between the two paths then is the minimum length dog leash needed to complete the walk, where both dog and owner are free to vary their walking speeds but are not allowed to backtrack. In our setting we compute the Frechet distance between two discrete paths $P : \{1, ..., L_p\} \to \mathbb{R}^3$, $Q : \{1, ..., L_q\} \to \mathbb{R}^3$ as defined in Eiter and Mannila[44]. In this definition, a coupling between $P$ and $Q$ is defined as a sequence of ordered pairs:

$$(P[a_1], Q[b_1]), (P[a_2], Q[b_2]), ..., (P[a_K], Q[b_K])$$

where the following conditions are put on $\{a_k\}, \{b_k\}$ to ensure that they enumerate through the whole sequences $P$ and $Q$:

- $a_1, b_1 = 1$
- $a_N = L_p$, $b_N = L_q$
- $a_k = a_{k-1}$ or $a_k = a_{k-1} + 1$
- $b_k = b_{k-1}$ or $b_k = b_{k-1} + 1$

Then the discrete Frechet distance is defined as:

$$\delta_{dF}(P, Q) = \min_{\text{coupling } \{a_k\}, \{b_k\}} \max_{k \in \{1, ..., K\}} |P[a_k] - Q[b_k]|$$

We use the standard Euclidean norm for $|\cdot|$. The discrete Frechet distance is an upper bound to the continuous Frechet distance between polygonal curves, and it can be computed more efficiently. Further, if we take a discrete Frechet distance of zero to be an equivalence relation, then $\delta_{dF}$ is a metric on this set of equivalence classes and thus is a natural way to compare non-branching neuronal reconstructions. In this work, all reconstruction are sampled at at least one point per micron.

Various other performance metrics have been proposed, including an arc-length based precision and recall[6], a critical node matching based Miss-Extra-Scores (MES)[45] and a vertex matching based spatial distance (SD)[3]. We chose to compute SD since it gives a picture of the average spatial distance between two reconstructions. This complements the Frechet distance described earlier, which computes the maximum spatial distance between two reconstructions.

The first step in computing the SD from reconstruction $P$ to reconstruction $Q$ is, for each point in $P$, finding the distance to the closest point in $Q$. Directed divergence (DDIV) of $P$ from $Q$ is then defined as the average of all these distances. Then, SD is computed by averaging the DDIV from $P$ to $Q$ and the DDIV from $Q$ to $P$.

**Statistics and reproducibility**. The first statistical analysis in this work pertains to foreground/background intensity distributions, and is encapsulated in Fig. 3. The figure uses data from three image subvolumes, which are provided in the Supplemental Data. In panel a, each curve is computed from 5000 randomly selected voxels in the image subvolume, and each error bar represents a single standard deviation of the Fisher z-transformation of the correlation coefficient. In panel b, each curve depicts a Gaussian kernel estimator that was fit to 5000 random voxels. The scipy implementation of Gaussian kernel density estimator was used, with Scott's rule to determine the bandwidth[38,39]. We provide the code to reproduce this figure in brainlit. Note that since the voxels are randomly chosen, the notebook may not reproduce the exact curves in the figure, but the trends, and our conclusions, are robust to different random samples.

The second statistical analysis in this work pertains to the reconstruction outcomes of the different algorithms, as shown in Fig. 7. A reconstruction is considered a Failure if the trace follows the axon for less than half its length. A reconstruction is considered a Partial Success if it follows the axon for at least half its length, but clearly deviates onto another structure at some point. Reconstructions that follow the axon from the start point to the soma are considered Successes. We compared success rates between algorithms using a two proportion z-test with a significance threshold of 0.05. We also documented the accuracy metrics (Frechet distance, spatial distance) of the successful traces. The outcome counts, the metrics, and the code to produce the plots are provided in the software package. Further, we provide the code that was used to run the ViterBrain reconstructions, and measure the accuracy metrics.

**Reporting summary**. Further information on research design is available in the Nature Research Reporting Summary linked to this article.

## Data availability

The datasets analyzed for this study are available in the Open Neurodata AWS account, https://registry.opendata.aws/open-neurodata/. Our package, brainlit provides examples of accessing this data.

## Code availability

The code used in this work is available in our open-source Python package brainlit: http://brainlit.neurodata.io/, and a tutorial on how to use the code is located at: http://brainlit.neurodata.io/viterbrain.html. The version associated with this paper is https://doi.org/10.5281/zenodo.6323454.

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

## Acknowledgements

This work is supported by the National Institutes of Health grant RF1MH121539, P41EB015909, R01NS086888, U19AG033655, the National Science Foundation Grant 2014862, and the National Institute of General Medical Sciences Grant T32GM119998. We thank the MouseLight team at HHMI Janelia for providing us with access to this data, and answering our questions about it.

## Author contributions

M.I.M. helped to develop the HMM and probabilistic model and D.J.T. advised on the theoretical direction of the manuscript. U.M. coordinated the data acquisition for the experiments. J.T.V. advised on the software design. T.L.A. designed the study, implemented the software, and managed the manuscript text/figures. All authors contributed to manuscript revision.

## Competing interests

M.I.M. owns a significant share of Anatomy Works with the arrangement being managed by Johns Hopkins University in accordance with its conflict of interest policies. The remaining authors declare that the research was conducted in the absence of any commercial or financial relationships that could be construed as a potential conflict of interest. The funders had no role in study design, data collection and analysis, decision to publish, or preparation of the manuscript.
