## [Peer Review File · Communications Biology]

Reviewers' comments:

Reviewer #1 (Remarks to the Author):

This paper presents a probabilistic model, named as ViterBrain, for neuron reconstruction. The authors integrate a newly designed hidden Markov model that directly encodes the neuron geometry with appearance models of neuron fluorescence images. The proposed idea sounds interesting and the research topic of better reconstructing the neuron paths is of high interest to the neuroscience community. While the backbone of the developed methodology seems convincing, the current manuscript has a lot of room for improvement. Below are my major comments and suggestions:

*While the structure of this paper is well organized, many technical details are sloppily defined or written; hence difficult for readers to follow. Some of the math notations (especially in section 4.2) are either missing, or defined after being used in equations (e.g., the notations of α_0 , α_1 , α_k).

*Not sure whether it is appropriate to call the definition of the 'most probable solution' as a 'Theorem'. I would at least expect a rigorous mathematical proof of how/why the proposed formulation can achieve a 'most probable solution'.

*For the estimation of foreground-background intensity distributions, I am wondering whether the authors have thoroughly validated the reliability/consistency of fitting Gaussian kernels to subsampled datasets. I assume different estimates of these intensity distributions will lead to different generations of fragments.

* The validation of the proposed approach is less convincing for two major reasons. First, the authors only select two baseline algorithms (APP2 and snake) for comparison out of twenty-six. Besides, the selected two baseline methods are not able to provide sufficient number of metrics for a fair comparison in Fig. 8. Second, it is not clear whether the results of all methods (on either spatial distance metrics or Frechet distance metrics) are statistically significantly different. A thorough statistical analysis would be necessary.

*What are the variations of different metrics in Fig. 8?

Minor comments:

*The segmentation of the neuron node in Fig. 1 is surprisingly off. I would expect a much better quality since the boundary of the node is quite clear. As the proposed model heavily depends on the quality of binary segmentation masks, I'd suggest the authors use better examples in Fig. 1.

*It would be helpful to show the original images of a) and b) in Fig. 7 without being overlaid by any manual labels or estimated reconstructions.

*It would be good to add a brief description of the tested dataset (e.g., the number and the resolution of images, etc.) in the result section.

Reviewer #2 (Remarks to the Author):

Automatic neuron reconstruction is a challenging task in brain clearing and imaging. Although many automatic reconstruction algorithms have been proposed over the past decade, most focus on single neuron images. This manuscript presents a probabilistic reconstruction method for multiple neuron images, which combines a hidden Markov process encoding neuron geometry with a random field appearance model of neuron fluorescence. Moreover, the proposed method, ViterBrain, performed better than two selected baselines in imperfect image segmentation. It is also noteworthy that the authors of this manuscript shared their implementation code on the Internet. However, a few major or minor issues are listed as follows.

1. What is the difference between this manuscript and an online reprint (arXiv:2106.02701) authored by the same authors? Perhaps, the authors should mention the reprint in this manuscript.
2. The organization of Section 2 and Section 4 can be improved. Specifically, Subsection 2.1 could be moved to the beginning of Section 4 because it only presents the overview of ViterBrain and does not provide any actual result. Besides, since Figure 2 is relatively simple, the authors should describe how the ViterBrain components interact with other modules to perform a given task.
3. What is the correlation of ViterBrain with image intensity modeling? For the reviewer's part, Subsection 2.2 appears to be less important than the following three ones in Section 2. The authors should make the motivation of this result more precise.
4. There are 26 reconstruction algorithms available in Vaa3D (version 3.2), and the authors selected two baselines to compare with the proposed method. However, the two baselines were proposed many years ago. Could this manuscript introduce more recently published methods, primarily based on deep learning?
5. The comparison among three reconstruction algorithms was conducted on a dataset of 10 partial axon reconstructions. For the reviewer's part, the experimental result is less convincing. Therefore, the authors should compare them on a larger-size dataset to demonstrate the effectiveness of the proposed method.
6. Two distance metrics are employed to measure the discrepancies between reconstruction algorithms and manual traces. Obviously, they are used at the level of individual examples. Could the authors provide additional metrics or statistical analysis methods for the whole dataset?
7. Hidden Markov modeling is a relatively old technique in computer science. Due to some inherent disadvantages, this technique has gradually been replaced by the recurrent neural network (RNN) architecture (more specifically, long short-term memory networks and gate recurrent units) in natural language processing and other sequential tasks. Therefore, the novelty of ViterBrain remains unknown unless the authors can demonstrate that the proposed method outperforms other RNN-based baselines.
8. Some recent papers could be further analyzed and discussed in the revised version of this manuscript.

[1] Hang Zhou, Shiwei Li, Anan Li, Qing Huang, Feng Xiong, Ning Li, Jiacheng Han, Hongtao Kang, Yijun Chen, Yun Li, Huimin Lin, Yu-Hui Zhang, Xiaohua Lv, Xiuli Liu, Hui Gong, Qingming Luo, Shaoqun Zeng, Tingwei Quan: GTree: an Open-source Tool for Dense Reconstruction of Brain-wide Neuronal Population. *Neuroinformatics* 19(2): 305-317 (2021)

[2] Heng Wang, Chaoyi Zhang, Jianhui Yu, Yang Song, Siqi Liu, Wojciech Chrzanowski, Weidong Cai: Voxel-Wise Cross-Volume Representation Learning for 3D Neuron Reconstruction. *Proc. MLMI@MICCAI 2021*: 248-257

[3] Shiwei Li, Tingwei Quan, Hang Zhou, Qing Huang, Tao Guan, Yijun Chen, Cheng Xu, Hongtao Kang, Anan Li, Ling Fu, Qingming Luo, Hui Gong, Shaoqun Zeng: Brain-Wide Shape Reconstruction of a Traced Neuron Using the Convex Image Segmentation Method. *Neuroinformatics* 18(2): 199-218 (2020)

[4] Qiufu Li, Linlin Shen: 3D Neuron Reconstruction in Tangled Neuronal Image With Deep Networks. *IEEE Trans. Medical Imaging* 39(2): 425-435 (2020)

[5] Miroslav Radojevic, Erik Meijering: Automated Neuron Reconstruction from 3D Fluorescence Microscopy Images Using Sequential Monte Carlo Estimation. *Neuroinformatics* 17(3): 423-442 (2019)

[6] Min Liu, Huiqiong Luo, Yinghui Tan, Xueping Wang, Weixun Chen: Improved V-Net Based Image Segmentation for 3D Neuron Reconstruction. *Proc. IEEE BIBM 2018*: 443-448

Department of Biomedical Engineering
3400 N. Charles Street / Wyman Park Building / Suite 400W
Baltimore, MD 21218
410-516-8120
mim@jhu.edu

Michael I. Miller, Ph.D.
Bessie Darling Massey Professor and Department Director

January 24, 2021

Dear Reviewers,

We truly appreciate the time and effort you invested in reading our submission and giving constructive feedback. We tried to address all the suggestions. Notably, we expanded our comparison to state of the art by using more algorithms, a larger dataset, and a more thorough statistical analysis of the results. We also overhauled several sections in the Results and Methods in order to make the writing clearer and more concise. Below is a point-by-point reproduction of your review comments, each followed by our corresponding responses.

Thank you for considering our updated manuscript.

Sincerely,

Michael I. Miller
Bessie Darling Massey Professor and Director, Department of Biomedical Engineering
Co-Director, Kavli Neuroscience Discovery Institute
Johns Hopkins University School of Medicine and Whiting School of Engineering

Reviewer #1 (Remarks to the Author):

This paper presents a probabilistic model, named as ViterBrain, for neuron reconstruction. The authors integrate a newly designed hidden Markov model that directly encodes the neuron geometry with appearance models of neuron fluorescence images. The proposed idea sounds interesting and the research topic of better reconstructing the neuron paths is of high interest to the neuroscience community. While the backbone of the developed methodology seems convincing, the current manuscript has a lot of room for improvement. Below are my major comments and suggestions:

- While the structure of this paper is well organized, many technical details are sloppily defined or written; hence difficult for readers to follow. Some of the math notations (especially in section 4.2) are either missing, or defined after being used in equations (e.g., the notations of $\alpha_0, \alpha_1, \alpha_k$).
 - We overhauled our description of the methods. In particular we:
 - Shortened our discussion and notation of the Poisson process underlying the imaging.
 - Moved all proofs to the supplement, but retained descriptions of the ideas behind the proofs.
 - Introduced the important variables ($\alpha_0, \alpha_1, \alpha_k, \alpha_d$) earlier, in the Results section.
 - From Section 2.2: *For that reason, we exploit the independent increments properties of Poisson emission conditioned on the underlying intensity model, but do not assume that the marginal probabilities are Poisson (or Gaussian), instead, we estimate the intensity distributions from the data itself using KDEs (denoted $\alpha_0(\cdot), \alpha_1(\cdot)$ in Section 4.1).*
 - From Section 2.3: *Our geometric prior has two hyperparameters, α_d and α_κ , which determine the influence of distance and curvature, respectively, on the probability of connection between two neuronal fragments.*
 - Cleaned up writing style.
- Not sure whether it is appropriate to call the definition of the ‘most probable solution’ as a ‘Theorem’. I would at least expect a rigorous mathematical proof of how/why the proposed formulation can achieve a ‘most probable solution’.
 - We combined it with the “Proposition” and renamed it “Statement” (Statement 1 in Section 4.3). The rigorous proof is in Supplement section S3 and is based on a recursive factoring.
- For the estimation of foreground-background intensity distributions, I am wondering whether the authors have thoroughly validated the reliability/consistency of fitting Gaussian kernels to subsampled datasets. I assume different estimates of these intensity distributions will lead to different generations of fragments.
 - Though our validation was mostly qualitative, we use a method that has been heavily studied in the applied mathematics literature. In particular, the method has been proven to be consistent in estimating arbitrary distributions (under some regularity

conditions). We added some details of the theoretical backing of our approach in section 4.1.

- From Section 4.1: *Despite the Poisson nature of the image acquisition process, simple scaling or shifting of the imaging data would mean the image intensities are no longer Poisson. To accommodate this effect, we estimate the foreground-background intensity distributions ($\alpha_1(\cdot)$ and $\alpha_0(\cdot)$) respectively) nonparametrically. The simplest nonparametric density estimation technique is using histograms. However, it can be difficult to choose the origin and bin width of histograms, so we opt for a kernel density estimate (KDE) approach. We estimate $\alpha_0(\cdot)$, $\alpha_1(\cdot)$ by labeling a subset of the data as foreground/background then fitting Gaussian KDEs to the labeled data (see Figure 3b). We use the scipy implementation of Gaussian KDEs [Virtanen et. al., 2020], with Scott's rule to determine the bandwidth parameter [Scott, 2015]. Under some assumptions on the derivatives of the underlying density, our approach converges to the true density as the number of samples increases (Theorem 6.1 in Scott [2015]). Further, the Scott's rule choice for bandwidth is (approximately) optimal with respect to mean integrated square error.*
- Our KDE's do not affect fragment generation (that is done by the Ilastik binary segmentation). The KDE's are instead used to compute the likelihood of the observed image data under a given neuronal trajectory. In other words, it does not affect the fragments, but could affect which sequence of fragments is chosen. Based on our method's performance on a variety of subvolumes, we are content with this approach.
- The validation of the proposed approach is less convincing for two major reasons. First, the authors only select two baseline algorithms (APP2 and snake) for comparison out of twenty-six. Besides, the selected two baseline methods are not able to provide sufficient number of metrics for a fair comparison in Fig. 8. Second, it is not clear whether the results of all methods (on either spatial distance metrics or Frechet distance metrics) are statistically significantly different. A thorough statistical analysis would be necessary.
 - We have significantly expanded our reconstruction experiment. Now, we have chosen four other baseline algorithms, APP2, GTree, Advantra, and Snake. There are indeed more algorithms in Vaa3d, but they are typically not thoroughly described (i.e. there is no accompanying publication that the reader can explore for troubleshooting purposes etc.), nor are they used or cited as frequently in field as the ones we chose, so we decided to limit our collection of algorithms to 5.
 - We more than triple the testing dataset – from 10 to 35 images. We also plot box and whisker plots of accuracy metrics of the reconstructions that are successful. We exclude the metrics of failed reconstruction because they would obscure the informative metrics.
 - See Figure 7
 - We also perform 2 proportion z-tests to statistically compare the success rates of the different algorithms.
 - From Section 2.5: *According to two proportion z-tests the success rate of ViterBrain (11/35) was higher than all other methods at $\alpha=0.05$. Also, APP2 had a higher success rate (4/35) than Advantra at $\alpha=0.05$.*
- What are the variations of different metrics in Fig. 8?

- The new box and whisker plots of the different metrics should adequately convey average and variation of these different values

Minor comments:

- The segmentation of the neuron node in Fig. 1 is surprisingly off. I would expect a much better quality since the boundary of the node is quite clear. As the proposed model heavily depends on the quality of binary segmentation masks, I'd suggest the authors use better examples in Fig. 1.
 - We modified Figure 1 to show a better segmentation. There are indeed still some visually obvious false negatives, which could be addressed with a less conservative binarization threshold. However, we maintain the conservative threshold for two reasons. First, a less conservative threshold would lead to more false positives in the background, and it would fuse together different neuronal processes (there is of course always a tradeoff). Second, we think it is more illustrative of our approach, which strings together distinct fragments. Our primary goal in Figure 1 is to describe a problem, and our approach.
- It would be helpful to show the original images of a) and b) in Fig. 7 without being overlaid by any manual labels or estimated reconstructions.
 - We added the original image to figure 7b, and now it is in the supplement as figure s2. We removed figure 7a after some rearrangement in order to satisfy the figure limit.
- It would be good to add a brief description of the tested dataset (e.g., the number and the resolution of images, etc.) in the result section.
 - We added more information (e.g. resolution, and number of images) about the dataset in sections 2.1 and 2.5.
 - From Section 2.1: *We validated our algorithm on subvolumes of one of the MouseLight whole-brain images [Winnubst et al., 2019]. The image was acquired via serial two-photon tomography at a resolution of 0.3 μm x 0.3 μm x 1 μm per voxel.*
 - From Section 2.5: *Each subvolume contains a cell body, and the initial part of its axon that is covered by the first ten points of the Janelia reconstruction. So, the subvolumes vary in size but usually encompass around 10^6 cubic microns.*

Reviewer #2 (Remarks to the Author):

Automatic neuron reconstruction is a challenging task in brain clearing and imaging. Although many automatic reconstruction algorithms have been proposed over the past decade, most focus on single neuron images. This manuscript presents a probabilistic reconstruction method for multiple neuron images, which combines a hidden Markov process encoding neuron geometry with a random field appearance model of neuron fluorescence. Moreover, the proposed method, ViterBrain, performed better than two selected baselines in imperfect image segmentation. It is also noteworthy that the authors of this manuscript shared their implementation code on the Internet. However, a few major or minor issues are listed as follows.

- What is the difference between this manuscript and an online reprint (arXiv:2106.02701) authored by the same authors? Perhaps, the authors should mention the reprint in this manuscript.
 - Yes, that arxiv post is a preprint version of this same article. Earlier, the arxiv version was not completely up to date, which may have caused some confusion. We have since updated that post to match this submitted version.
- The organization of Section 2 and Section 4 can be improved. Specifically, Subsection 2.1 could be moved to the beginning of Section 4 because it only presents the overview of ViterBrain and does not provide any actual result. Besides, since Figure 2 is relatively simple, the authors should describe how the ViterBrain components interact with other modules to perform a given task.
 - Regarding section 2.1, we are inclined to keep it there since it describes the primary result of our work (the ViterBrain algorithm) and we also think it is important as background for the following sections. However, in order to make it more appropriate for the Results section, we tweaked its contents (e.g. added link to our Python package). This structure of starting the results with an overview is in part inspired by other papers in the same field, such as the G-Cut paper (Li et. Al. 2019 Nature Communications).
 - We added some of the important equations and notation to Figure 2 in order to more closely connect the figure with the notation in the Results/Methods sections.
- What is the correlation of ViterBrain with image intensity modeling? For the reviewer's part, Subsection 2.2 appears to be less important than the following three ones in Section 2. The authors should make the motivation of this result more precise.
 - We worked on the clarity and precision of the writing in this section. The first important takeaway from this section is the evidence it gives for our conditional independence assumption of the observed image. Our model assumes that, for example, two different voxels that are already known to be background have independent image intensities. We added this motivation to section 2.2.
 - From Section 2.2: *Figure 3a shows the correlations of image intensities between voxels at varying distances of separation. As is typical for natural images, voxels that are close by each other have positively correlated intensities, and those farther away are uncorrelated. In the case of foreground voxels, correlations become weak beyond a distance of about 10 microns, with background voxel correlation decaying rapidly. This lends support to our assumption that voxel intensities are conditionally independent processes, conditioned on the foreground/background model (Eq. 3b). This assumption is one of the central features of our model because it provides for computational tractability.*

- The second important takeaway is that the conditional distributions do not appear Gaussian or Poisson, which supports our nonparametric approach using KDEs. We added this motivation to section 2.2 as well.
 - From Section 2.2: *Figure 3b shows kernel density estimates (KDEs) of the foreground and background image intensity distributions. The distributions vary greatly between the three image subvolumes, implying that modeling the image process as homogeneous throughout the whole brain would be inappropriate. Additionally, the distributions do not appear to be either Gaussian or Poisson. Indeed, Kolmogorov-Smirnov tests rejected the null hypothesis for both Gaussian and Poisson goodness of fit in all cases, with all p-values below 10^{-16} . For that reason, we exploit the independent increments properties of Poisson emission conditioned on the underlying intensity model, but do not assume that the marginal probabilities are Poisson (or Gaussian), instead, we estimate the intensity distributions from the data itself using KDEs (denoted $\delta_{\alpha_0(\cdot)}, \alpha_1(\cdot)$ in Section 4.1)*
- There are 26 reconstruction algorithms available in Vaa3D (version 3.2), and the authors selected two baselines to compare with the proposed method. However, the two baselines were proposed many years ago. Could this manuscript introduce more recently published methods, primarily based on deep learning?
 - We added some more recently published methods, including and Advantra (Radojevic et. al. 2017) and GTree (Zhou 2020). Much of the deep learning literature in neuron reconstruction focuses on image preprocessing, or image segmentation (e.g. Wang et. al. 2021, Li et. al. 2020, Liu et. al. 2018), rather than the reconstruction problem itself. However we reference these methods in the Discussion as possible methods to supplement or improve our algorithm.
- The comparison among three reconstruction algorithms was conducted on a dataset of 10 partial axon reconstructions. For the reviewer's part, the experimental result is less convincing. Therefore, the authors should compare them on a larger-size dataset to demonstrate the effectiveness of the proposed method.
 - We expanded the dataset to 35 partial axons.
- Two distance metrics are employed to measure the discrepancies between reconstruction algorithms and manual traces. Obviously, they are used at the level of individual examples. Could the authors provide additional metrics or statistical analysis methods for the whole dataset?
 - We performed two-proportion z-tests to compare success rates between algorithms. Additionally, we produced box and whisker plots of the two distance metrics for a more descriptive analysis of the algorithms' performances.
 - See Figure 7
- Hidden Markov modeling is a relatively old technique in computer science. Due to some inherent disadvantages, this technique has gradually been replaced by the recurrent neural network (RNN) architecture (more specifically, long short-term memory networks and gate recurrent units) in natural language processing and other sequential tasks. Therefore, the novelty of ViterBrain remains unknown unless the authors can demonstrate that the proposed method outperforms other RNN-based baselines.
 - It is true that RNN approaches have made tremendous strides in many sequential decision processes, which had been classically solved by HMMs. However, we could not

find any notable publications about applying RNNs approaches to neuron reconstruction. We added a sentence to the discussion saying that this would be a ripe avenue for future work.

- From Section 3: *Future benchmark comparisons could include reinforcement learning, or recurrent neural network approaches, which have become prevalent in sequential decision processes. However, there is not much scientific literature on these approaches to neuron reconstruction with accompanying functional code.*
 - We do, however, reference a deep reinforcement learning method (Dai et. al. 2019). Unfortunately, the software accompanying this publication has issues (when we were following their readme, there is an error while running the training script), so we did not add it to our algorithm comparison.
- **Some recent papers could be further analyzed and discussed in the revised version of this manuscript.**
 - These references you gave helped expand our Introduction and Discussion sections. We mention Wang et. al. 2021, Liu et. al. 2018, and Li et. al. 2020 as deep learning image segmentation methods which could be used to generate the inputs to ViterBrain. We added Zhou et. al. and Radojevic et. al. to our algorithm comparisons (GTree and Advantra respectively). Lastly, we mention Li et. al. 2020 as method which could transform the reconstructions from ViterBrain into complete image segmentations.
 - From Section 3: *We chose Ilastik to generate image masks because of its convenient graphical user interface, and high performance on a small number of samples [Bergs et al., 2019]. However, masks could also be generated using a deep learning based model such as Liu et al. [2018], Li and Shen [2019] or Wang et al. [2021].*
 - From Section 3: *The traces generated by our pipeline could also be paired with tools like the one presented in Li et al. [2020] which can turn traces into full neuron segmentations complete with axon and dendrite thickness measurements.*

[1] Hang Zhou, Shiwei Li, Anan Li, Qing Huang, Feng Xiong, Ning Li, Jiacheng Han, Hongtao Kang, Yijun Chen, Yun Li, Huimin Lin, Yu-Hui Zhang, Xiaohua Lv, Xiuli Liu, Hui Gong, Qingming Luo, Shaoqun Zeng, Tingwei Quan: GTree: an Open-source Tool for Dense Reconstruction of Brain-wide Neuronal Population. *Neuroinformatics* 19(2): 305-317 (2021)

[2] Heng Wang, Chaoyi Zhang, Jianhui Yu, Yang Song, Siqi Liu, Wojciech Chrzanowski, Weidong Cai: Voxel-Wise Cross-Volume Representation Learning for 3D Neuron Reconstruction. *Proc. MLMI@MICCAI 2021*: 248-257

[3] Shiwei Li, Tingwei Quan, Hang Zhou, Qing Huang, Tao Guan, Yijun Chen, Cheng Xu, Hongtao Kang, Anan Li, Ling Fu, Qingming Luo, Hui Gong, Shaoqun Zeng: Brain-Wide Shape Reconstruction of a Traced Neuron Using the Convex Image Segmentation Method. *Neuroinformatics* 18(2): 199-218 (2020)

[4] Qiufu Li, Linlin Shen: 3D Neuron Reconstruction in Tangled Neuronal Image With Deep Networks. *IEEE Trans. Medical Imaging* 39(2): 425-435 (2020)

[5] Miroslav Radojevic, Erik Meijering: Automated Neuron Reconstruction from 3D Fluorescence Microscopy Images Using Sequential Monte Carlo Estimation. *Neuroinformatics* 17(3): 423-442 (2019)

[6] Min Liu, Huiqiong Luo, Yinghui Tan, Xueping Wang, Weixun Chen: Improved V-Net Based Image Segmentation for 3D Neuron Reconstruction. Proc. IEEE BIBM 2018: 443-448

REVIEWERS' COMMENTS:

Reviewer #1:

Remarks to the Author:

The authors have well addressed all my concerns. I have no further comments on this manuscript.

Reviewer #2:

Remarks to the Author:

The authors have answered the questions raised by the reviewers and made significant improvements on the quality of the manuscript. Therefore, it can be accepted for publication in the journal.